# AerialExtreMatch: A Benchmark for Extreme-View Image Matching and Localization

## Abstract

Image matching serves as a core component for UAV localization guided by satellite imagery. However, this task remains highly challenging due to the extreme viewpoint discrepancies between low-altitude UAV images and nadir-view satellite maps. Existing datasets primarily focus on ground-level or high-altitude UAV imagery, lacking sufficient coverage of the geometric variations typical of real aerial scenarios. To address these limitations, we introduce **AerialExtreMatch**, a large-scale, high-fidelity dataset tailored for extreme-view image matching and UAV localization. It consists of approximately 1.5 million synthetic image pairs rendered from high-quality 3D scenes, simulating diverse UAV and satellite viewpoints to enable robust training of image matching models. To support fine-grained evaluation, we construct a hierarchical benchmark with 32 difficulty levels. These are defined using three geometric criteria: overlap ratio, scale variation, and pitch difference. In addition, we collect a real-world UAV localization dataset with geo-aligned reference maps of varying visual quality. Extensive experiments involving 16 representative detector-based and detector-free methods demonstrate that models trained on AerialExtreMatch achieve substantial performance gains in both image matching and real-world localization under extreme-view conditions. The dataset and code will be released upon acceptance.

## 1 Introduction

Image matching has become a pivotal technique for satellite-guided visual localization of low-altitude unmanned aerial vehicles (UAVs). Accurate pose estimation is particularly critical for mission-oriented applications such as field rescue Bejiga et al. (2017); Silvagni et al. (2017) and large-scale scene reconstruction Maboudi et al. (2023). While UAV-based reconstruction models have been commonly employed for these tasks, satellite-based reconstructions offer distinct advantages, including rapid updatability and scalability over vast geographic regions. Nevertheless, a fundamental challenge arises from the inherent disparity in imaging perspectives: satellite imagery is acquired from high-altitude orbits with *top-down orthographic* views, whereas low-altitude UAVs typically capture *oblique* images. This pronounced viewpoint difference greatly hinders the establishment of accurate feature correspondences, thus limiting the robustness and reliability of existing image matching methods.

Contemporary image matching methods Shen et al. (2024); Leroy et al. (2024) are predominantly trained and evaluated on ground-level datasets, which inherently lack the angular variations present in aerial scenarios. This dataset bias substantially limits its generalizability and effectiveness in aerial image matching tasks. Although the rapid growth of the low-altitude UAV has motivated recent efforts to construct aerial localization benchmarks from reconstructed real-world scenes Wu et al. (2024); Chen et al. (2025); Ye et al. (2025), the high cost of data acquisition remains a significant barrier to the large-scale deployment of these approaches.

To overcome the performance limitations caused by the scarcity of aerial datasets and to establish a fair benchmark for algorithm evaluation, we propose a method for generating realistic synthetic data by leveraging high-fidelity 3D models and advanced rendering techniques. Our approach not only substantially reduces the cost of data collection but also inherently preserves privacy. Specifically, we employ a diverse set of 3D scene models provided by Cesium for Unreal, in combination with Unreal Engine 5 and AirSim, to systematically simulate both UAV aerial viewpoints and satellite overhead

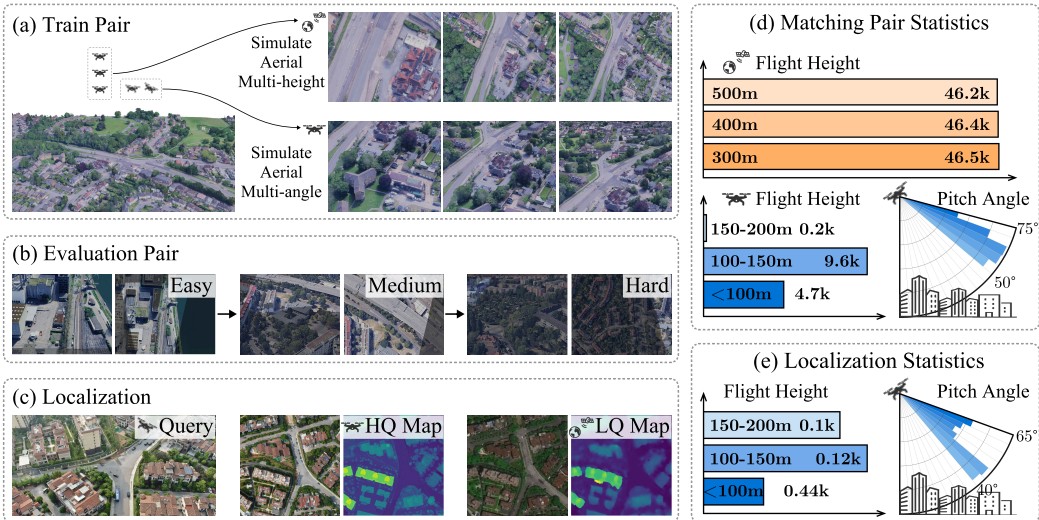

Figure 1: Overview of the proposed **AerialExtreMatch**, which consists of three components: **Train Pair**, **Evaluation Pair**, and **Localization**. The **Train Pair** set include approximately 1.5M pairs with variations in flight altitude and pitch for supporting model train. The **Evaluation Pair** set defines 32 difficulty levels, ranging from easy to hard, to enable fine-grained performance analysis. Invisibility is occluded with a black mask. The **Localization** component evaluates pose accuracy using real UAV query images and two geo-aligned reference maps with varying visual quality: a high-quality map reconstructed from UAV imagery (HQ) and a low-quality map derived from satellite data (LQ).

perspectives. This framework allows us to construct a comprehensive training dataset tailored for extreme-view image matching tasks. To effectively capture multi-scale geometric transformations, we generate multiple images at varying altitudes for each viewpoint within a given scene.

Compared to real-world data, synthetic data provides greater flexibility in simulating diverse viewpoints and offers precise control over the conditions of data generation. Leveraging these advantages, we introduce a fine-grained evaluation framework that facilitates comprehensive analysis of image matching robustness under varying geometric configurations. Specifically, we propose the *Evaluation Pair*, a structured 32-level hierarchy of matching difficulty defined by three key geometric criteria: overlap ratio, pitch difference, and scale.

Table 1: **Overview of existing image matching benchmark and UAV localization datasets.**

| Dataset | Type | Depth | Viewpoint Variation | Graded Evaluation | Supports Match Training |
|---|---|---|---|---|---|
| MegaDepth Li & Snavely (2018) | 📱 | ✓ | yaw | ✗ | ✓ |
| AerialMegaDepth Vuong et al. (2025) | ✈ | ✓ | pitch | ✗ | ✓ |
| ScanNet Dai et al. (2017) | 📱 | ✓ | yaw | ✗ | ✓ |
| ScanNet++ Yeshwanth et al. (2023) | 📱 | ✓ | yaw | ✗ | ✓ |
| HPatches Balntas et al. (2017) | 📱 | ✗ | yaw | ✗ | ✓ |
| BlendedMVS Yao et al. (2020) | 📱✈ | ✓ | yaw+pitch | ✗ | ✓ |
| Waymo Sun et al. (2020) | 🚗 | ✓ | yaw | ✗ | ✓ |
| RUBIK Loiseau & Bourmaud (2025) | 🚗 | ✗ | yaw | ✓ | ✗ |
| UAVD4L Wu et al. (2024) | ✈ | ✗ | pitch | ✗ | ✗ |
| AnyVisLoc Ye et al. (2025) | ✈ | ✗ | yaw | ✗ | ✗ |
| UAVVisLoc Xu et al. (2024) | ✈ | ✗ | pitch | ✗ | ✗ |
| **AerialExtreMatch (ours)** | ✈ | ✓ | pitch | ✓ | ✓ |

To further validate performance under real-world conditions, we compile geographically aligned reference maps with varying visual quality. These include digital surface models (DSMs) and digital orthophoto maps (DOMs) rendered from high-quality textured 3D models, as well as reconstructions derived from satellite imagery. The query set comprises real UAV-captured images acquired at low

altitudes, each paired with noisy prior poses and accurate ground-truth camera parameters. Together, these components constitute a comprehensive benchmark dataset for UAV-based visual localization. The complete pipeline for image matching and localization defines our proposed **AerialExtreMatch**, as illustrated in Figure 1.

We adopt RoMa Edstedt et al. (2024b), a state-of-the-art image matching method, as the baseline for training. Experimental results demonstrate that training on AerialExtreMatch significantly enhances its robustness to large viewpoint variations. In addition, we conduct a comprehensive evaluation involving 16 representative approaches, spanning both detector-based and detector-free methods. The results show that under favorable conditions—characterized by high overlap, small scale differences, and minimal pitch variation—all evaluated methods perform reliably. However, under extreme-view conditions with low overlap and large pitch differences, the performance of existing methods degrades significantly. Notably, the RoMa variant trained on our dataset achieves the best overall matching accuracy. In the real-world localization benchmark, it similarly attains superior performance, achieving recall rates on high-quality reference maps that are more than twice those on low-quality maps under the same evaluation protocol.

The contributions of this work can be summarized in three aspects:

- We construct a synthetic dataset for supporting aerial-view image matching. The evaluation pair is further organized into difficulty levels based on geometric variations.

- We collect a real-world dataset for low-altitude UAV localization, consisting of geo-aligned high-quality and low-quality reference maps.

- We provide a comprehensive benchmark involving 17 representative methods, demonstrating the effectiveness of the proposed dataset.

## 2 Related Work

### 2.1 Image Matching Benchmarks

**Training Data.** Large-scale RGB-depth training, exemplified by DUSt3R Wang et al. (2024b); Leroy et al. (2024); Wang et al. (2024a; 2025), has proven effective for depth estimation, pose regression, and scene reconstruction, with clear scalability benefits. Recent works such as MatchAnything He et al. (2025) and MINIMA Ren et al. (2025) extend this paradigm to image matching, where both data diversity and scale are critical. However, existing datasets Li & Snavely (2018); Dai et al. (2017) mainly cover ground-level scenes with limited viewpoint variation, lacking aerial-specific transformations.

**Evaluation Data.** HPatches Balntas et al. (2017) lacks depth and focuses on homography. MegaDepth Li & Snavely (2018) and ScanNet Dai et al. (2017) are widely used but do not stratify difficulty, limiting failure analysis. RUBIK Loiseau & Bourmaud (2025) quantifies difficulty via geometric transformations using nuScenes Caesar et al. (2020), but lacks ground-truth depth and is restricted to vehicle-centric urban scenes. In contrast, we design a benchmark for aerial viewpoint variation and introduce three geometric metrics to stratify difficulty.

### 2.2 Satellite-Guided UAV Localization Dataset

Localizing low-altitude UAVs with satellite imagery is difficult due to extreme viewpoint gaps. Existing datasets focus on high-altitude nadir views He et al. (2024); Keetha et al. (2023). UAVD4L Wu et al. (2024) includes multi-view low-altitude data but relies on costly oblique reconstructions. Other works Ye et al. (2025) use satellite-reconstructed maps, but with limited quality. These datasets remain small and based on real UAV data, thus unsuitable for training large models. We contribute a real-world localization dataset with multi-quality reference maps, and further synthesize large-scale aerial imagery from photorealistic 3D models. As shown in Table 1, no prior dataset jointly supports large-scale image matching and realistic UAV localization.

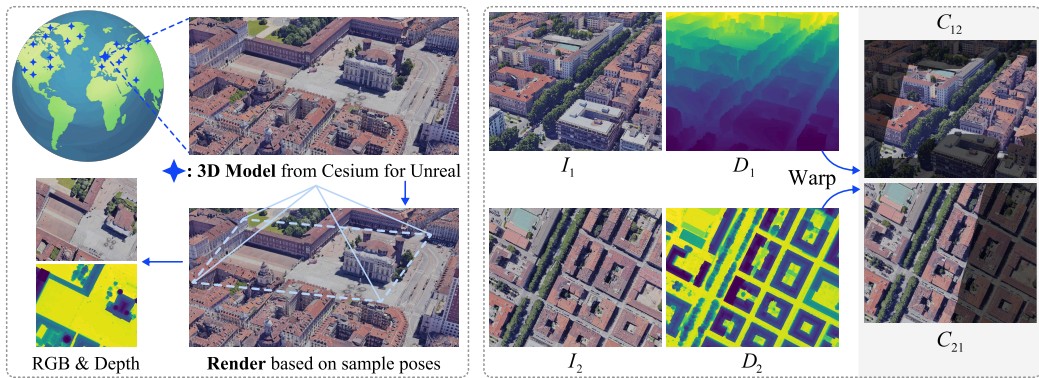

Figure 2: **Left:** *Training data generation.* Cesium for Unreal provides high-quality 3D models, enabling the rendering of RGB and depth images from diverse aerial viewpoints by sampling camera poses with varying altitudes and angles. **Right:** *Co-visibility estimation.* Given intrinsics $K$, extrinsics $P$, and RGB-depth image pairs $(I_1, D_1)$ and $(I_2, D_2)$, the co-visible masks $C_{12}$ and $C_{21}$ are computed by warping 3D points from one view to the other via geometric reprojection.

## 3   AERIALEXTREMATCH

**AerialExtreMatch** is a large-scale dataset tailored for training image matching, conducting hierarchical evaluations, and supporting real-world UAV localization tasks. Figure 2 presents the overall pipeline for collecting RGB-depth pairs and estimating the co-visibility mask. The dataset is constructed using high-quality 3D models and a photorealistic simulation engine. It comprises: (1) the generation of training data (Section 3.1); (2) the categorization protocol for match evaluation (Section 3.2); and (3) the design of the localization benchmark (Section 3.3).

### 3.1   TRAINING DATA COLLECTION

Most existing image matching datasets Li & Snavely (2018); Dai et al. (2017) are collected using hand-held devices, resulting in limited geometric diversity, particularly in variations in aerial viewpoints. This constraint significantly hampers the generalization ability of matching models. To address this limitation, we construct a dataset that incorporates diverse aerial viewpoint transformations to enhance model robustness.

Specifically, we employ Unreal Engine 5 with the AirSim plugin, in conjunction with l Cesium for Unreal, which provides high-quality 3D models, to generate paired RGB-depth images. From various urban and natural scenes available in Cesium for Unreal, we select 63 distinct regions and simulate both oblique UAV perspectives and nadir satellite views, yielding a training dataset of approximately 1.5M image pairs.

To better emulate real-world low-altitude UAV flight conditions, simulated UAV views are rendered at altitudes ranging from 50 m to 200 m, with pitch angles between $50°$ and $75°$. To introduce multi-scale geometric variation, we generate satellite-view at three heights—300 m, 400 m, and 500 m—at the same geographic location. Each RGB-depth pair is rendered at a resolution of $1280 \times 1024$ pixels. More details are provided in the appendix A.1.

### 3.2   GRADED EVALUATION GENERATION

The *Train Pair* and *Evaluation Pair* are both synthesized, where each image is associated with an RGB image $I$, a depth map $D$, and the corresponding intrinsics $K$ and extrinsics $P$. For evaluation, we compute geometric criteria between image pairs, including overlap ratio, pitch difference, and scale, to establish graded difficulty.

Given an image pair $(I_1, I_2)$ with known camera intrinsics $(K_1, K_2)$, extrinsics $(P_1, P_2)$, and depth maps $(D_1, D_2)$, we estimate co-visibility masks $C_{12}$ and $C_{21}$. For each pixel $\mathbf{p}_1 \in I_1$, the corresponding location in $I_2$ is obtained via:

$$\mathbf{p}_2 = \pi \left( K_2 \left( R_{21} \left( D_1(\mathbf{p}_1) \cdot K_1^{-1} \begin{bmatrix} \mathbf{p}_1 \\ 1 \end{bmatrix} \right) + \mathbf{t}_{21} \right) \right), \tag{1}$$

Where $R_{21}$ and $\mathbf{t}_{21}$ denote the relative rotation and translation from camera 1 to camera 2, and $\pi(\cdot)$ represents the perspective projection that normalizes by the depth coordinate.

The reprojected depth $D_2'(\mathbf{p}_2)$ is defined as the $z$-coordinate of the transformed 3D point:

$$D_2'(\mathbf{p}_2) = \left( R_{21} \left( D_1(\mathbf{p}_1) \cdot K_1^{-1} \begin{bmatrix} \mathbf{p}_1 \\ 1 \end{bmatrix} \right) + \mathbf{t}_{21} \right)^{(z)}, \tag{2}$$

Where $(\cdot)^{(z)}$ denotes extracting the third (depth) component of a 3D vector.

The visibility mask $C_{12}(\mathbf{p}_1)$ is computed as:

$$C_{12}(\mathbf{p}_1) = \begin{cases} 1, & \text{if } |D_2'(\mathbf{p}_2) - D_2(\mathbf{p}_2)| < \epsilon \cdot D_2(\mathbf{p}_2), \\ 0, & \text{otherwise,} \end{cases} \tag{3}$$

Where $\epsilon$ is empirically set to 0.05.

**Criteria.** Building upon the co-visibility masks, we introduce three geometric metrics to quantitatively assess the matching difficulty of each image pair.

**(1) Overlap Ratio:** Defined as the ratio of co-visible pixels between the two images to the average number of total pixels, and computed as:

$$\text{Overlap} = \frac{\sum_{\mathbf{p}_1} C_{12}(\mathbf{p}_1) + \sum_{\mathbf{p}_2} C_{21}(\mathbf{p}_2)}{|\mathbf{p}_1| + |\mathbf{p}_2|}, \tag{4}$$

where $|\mathbf{p}_1|$ and $|\mathbf{p}_2|$ denote the total number of pixels in $I_1$ and $I_2$, respectively. The visibility masks $C_{12}$ and $C_{21}$ represent pixel-level co-visibility from $I_1$ to $I_2$ and vice versa.

**(2) Pitch Difference:** Measures the absolute difference in pitch angles between the two cameras. One image simulates a nadir-view satellite perspective, while the other represents an oblique UAV view.

**(3) Scale:** Captures the scale difference between the two views based on the 2D coverage of 3D-projected corner points onto the $xy$-plane. Given the 3D corner points $\{\mathbf{x}_i^{(1)}\}_{i=1}^4$ and $\{\mathbf{x}_i^{(2)}\}_{i=1}^4$ for $I_1$ and $I_2$, the projected areas $A_1$ and $A_2$ are computed as:

$$A_k = \left( \max_i \mathbf{x}_i^{(k,x)} - \min_i \mathbf{x}_i^{(k,x)} \right) \times \left( \max_i \mathbf{x}_i^{(k,y)} - \min_i \mathbf{x}_i^{(k,y)} \right), \quad k \in \{1, 2\}, \tag{5}$$

where $\mathbf{x}_i^{(k,x)}$ and $\mathbf{x}_i^{(k,y)}$ denote the $x$- and $y$-coordinates of the $i$-th corner point in image $k$.

The scale ratio is then defined as:

$$\text{Scale} = \max \left( \frac{A_1}{A_2}, \frac{A_2}{A_1} \right).$$

**Benchmark Organization.** Based on the three criteria defined above, we construct the *Evaluation Pair* by categorizing image pairs into discrete difficulty levels. Examples are shown in Figure 3. For each pair with a co-visibility mask $C$, we compute the corresponding metrics and discretize them into the following bins:

- **Overlap Ratio** (four bins): $<20$, 20–40, 40–60, and $>60$;
- **Pitch Difference** (four bins): 50–55, 55–60, 60–65, and 65–70 degrees;
- **Scale Variation** (two bins): 1–2, and $>2$;

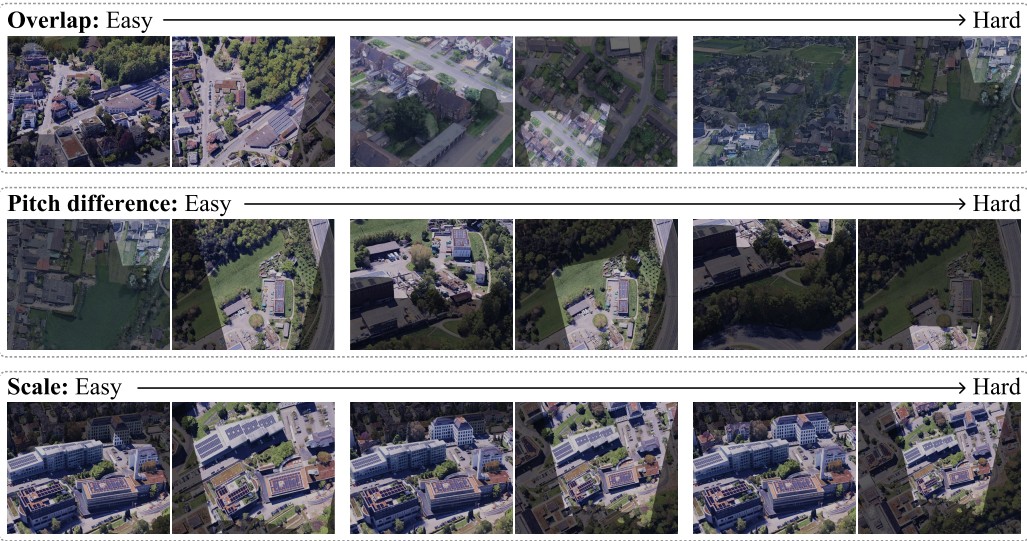

Figure 3: Visualization examples pairs under different difficulties.

We select up to 1k image pairs for each valid combination of difficulty bins. If fewer than 1k pairs are available, all available pairs are included. This sampling yields approximately 30k image pairs, broadly covering the geometric challenges commonly encountered in aerial image matching scenarios.

### 3.3 LOCALIZATION BENCHMARK

**Query Image Collection.** We collect query images using a DJI M300 RTK drone DJI300 equipped with a DJI H20T camera H20T. To simulate low-altitude, oblique flight conditions, both the drone's altitude and the camera's pitch angle are carefully controlled during data acquisition. Leveraging GPS priors, accurate camera poses are estimated using the Render2Loc localization method Yan et al. (2023). Details of the ground truth generation process are provided in the supplementary material.

**Reference Data Preparation.** Reference images are captured using a DJI M300 RTK drone DJI300 equipped with a professional five-lens camera, the SHARE PSDK 102S 102s. The aerial imagery is processed using modern 3D reconstruction techniques to generate a digital orthophoto map (DOM) and a digital surface model (DSM). In addition, satellite-derived DSM and DOM data covering the same geographic region are acquired from commercial providers and spatially aligned.

**Pair Construction.** Given the prior information of each query image, including camera intrinsics and noisy extrinsics, we project each reference 3D point onto the query image plane to determine the corresponding pixel location. Based on this projection, we extract a cropped patch from the reference DSM/DOM to form a localization image pair. The projection is defined by the following equation:

$$\mathbf{u} = \pi\left(\mathbf{K}\left(\mathbf{R}_{\text{aerial}}\mathbf{X}_{\text{sate}} + \mathbf{t}_{\text{aerial}}\right)\right), \quad \text{where } \pi = \left(\frac{x}{z}, \frac{y}{z}\right), \; z > 0, \; 0 \leq u \leq W_{\text{aerial}}, \; 0 \leq v \leq H_{\text{aerial}}. \tag{6}$$

## 4 EXPERIMENT

**Training Details.** We generate a total of ∼1.5M image pairs in **AerialExtreMatch**, comprising nadir-view and oblique-view images captured at varying altitudes. For training the matching model, we adopt the official implementation of RoMa Edstedt et al. (2024b), using a combination of MegaDepth Li & Snavely (2018) and AerialExtreMatch as the training data. The learning rates are

set to $6.25 \times 10^{-7}$ for the encoder and $1.25 \times 10^{-5}$ for the decoder, and we use AdamW Loshchilov & Hutter (2017) as the optimizer. Training is conducted on 8 NVIDIA 3090 GPUs (24GB) with a batchsize of 32, taking approximately two days to complete.

**Evaluation Methods.**   We compare our trained model against a total of 16 representative methods, including detector-based and detector-free approaches.

For detector-based methods, we select SuperPoint DeTone et al. (2018), DISK Tyszkiewicz et al. (2020), and ALIKED Zhao et al. (2023) as keypoint detectors, and employ LightGlue Lindenberger et al. (2023) and XFeat Potje et al. (2024) for feature matching. Additionally, the recent large geometry model VGGT Wang et al. (2025) provides image matching results by using ALIKED Zhao et al. (2023) for keypoint extraction and a tracking head for matching, as described in the original paper.

For detector-free methods, we evaluate LoFTR Sun et al. (2021), ELoFTR Wang et al. (2024c), ASpanFormer Chen et al. (2022), the official RoMa Edstedt et al. (2024b), RoMa trained with the GIM Shen et al. (2024) paradigm (denoted as **GIM**), RoMa trained in a cross-modal manner He et al. (2025) (denoted as **MA**), as well as DUSt3R Wang et al. (2024b) and MASt3R Leroy et al. (2024), which perform image matching via pointmap correspondence.

To ensure fair comparison, all experiments are conducted on the same hardware configuration: NVIDIA 3090 GPUs. For RoMa-based methods, all training and evaluation settings are kept identical except for the model checkpoint. Other methods are used with their default hyperparameters, and the maximum input image size is set such that the longer side does not exceed 1024.

## 4.1 MATCH BENCHMARK RESULTS

**Evaluation Protocol.**   Following Sun et al. (2021); DeTone et al. (2018), we report the AUC of pose errors at a threshold of $5°$, where the pose error is defined as the maximum of the rotation and translation errors. Camera poses are estimated by computing the fundamental matrix from the predicted correspondences using the RANSAC.

**Results.**   We present in Figure 4 the image matching results of RoMa-based methods, two detector-based, and one pointmap-based approach. The difficulty is structured cyclically every four levels: within each cycle, the overlap ratio remains constant, while increasing the level index corresponds to larger pitch angle differences. Across cycles, a higher level index indicates decreasing overlap.

Level 1 represents large overlap, small scale variation, and minimal pitch difference, both detector-based and detector-free methods achieve satisfactory performance. However, as the pitch difference increases within the same overlap setting, the performance of most methods deteriorates. Notably, in the most challenging condition (pitch difference of $70°$–$75°$), our trained RoMa consistently achieves the best results, outperforming the second-best method by a margin of 20% in the hardest level. As shown in the qualitative results, our trained RoMa-based model is the only one that predicts correct correspondences within co-visible regions. Other training paradigms produce numerous incorrect matches. Detector-based methods fail under such extreme conditions and generate error matches. MASt3R Leroy et al. (2024) benefits from training on aerial data and produces a few sparse matches in co-visible areas. More results are provided in the supplementary material.

## 4.2 LOCALIZATION BENCHMARK RESULTS

**Evaluation Protocol.**   We follow the standard visual localization evaluation protocol Sattler et al. (2018); Wu et al. (2024) and report the results under three commonly used thresholds: $(5\,\mathrm{m}, 1°)$, $(10\,\mathrm{m}, 1°)$, and $(20\,\mathrm{m}, 2°)$.

**Results.**   Table 2 reports the visual localization results of 17 methods on both the high-quality (HQ) and low-quality (LQ) reference maps. On the HQ map, most methods perform well. Among detector-based methods, the combination of ALIKED Zhao et al. (2023) and LightGlue Lindenberger et al. (2023) achieves the best performance. For detector-free approaches, our trained RoMa outperforms all other methods, achieving a recall of 97.35% at the $(5\mathrm{m}, 1°)$ threshold, 6.0% points higher than

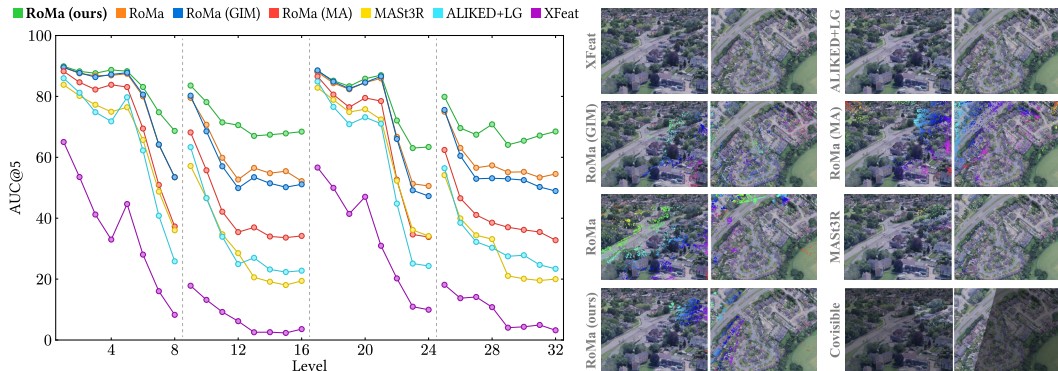

Figure 4: **Evaluation on AerialExtreMatch for image matching.** RoMa-based methods exhibit strong performance across various conditions, with our trained RoMa achieving the best results under challenging scenarios, including low overlap and large pitch differences. Each group of four levels shares the same overlap ratio, while the pitch difference gradually increases within each group as the level increases. Every 16 levels form a complete cycle across different scales. The right panel shows qualitative matching results under extreme conditions, where invisible regions are masked in black. (Zoom in for details; matching correspondences are indicated by same-colored points.)

Table 2: **Visual localization results on different quality maps.** The left panel shows results on the high-quality (HQ) map, while the right panel presents results on the low-quality (LQ) map. Our trained RoMa achieves superior performance across both settings, with the best results highlighted in **bold**.

| Method | (5m, 1°) ↑ | (10m, 1°) ↑ | (20m, 2°) ↑ | (5m, 1°) ↑ | (10m, 1°) ↑ | (20m, 2°) ↑ |
|---|---|---|---|---|---|---|
| ALIKED Zhao et al. (2023)+LG Lindenberger et al. (2023) | 86.74 | 86.74 | 87.12 | 3.03 | 5.68 | 14.39 |
| DISK Tyszkiewicz et al. (2020)+LG Lindenberger et al. (2023) | 82.58 | 82.58 | 83.71 | 0.00 | 0.00 | 0.00 |
| SP DeTone et al. (2018)+LG | 79.92 | 79.92 | 84.85 | 6.82 | 9.47 | 17.42 |
| DeDoDe Edstedt et al. (2024a) | 37.50 | 37.88 | 68.94 | 0.00 | 0.00 | 0.00 |
| XFeat Potje et al. (2024) | 54.92 | 54.92 | 74.62 | 0.38 | 0.38 | 1.14 |
| XFeat* Potje et al. (2024) | 56.06 | 56.44 | 73.49 | 0.76 | 0.00 | 3.79 |
| XFeat Potje et al. (2024) +LG Lindenberger et al. (2023) | 77.27 | 77.27 | 81.44 | 0.00 | 0.00 | 0.38 |
| ALIKED Zhao et al. (2023)+VGGT Wang et al. (2025) | 2.65 | 4.55 | 30.30 | 0.00 | 0.00 | 0.00 |
| LoFTR Sun et al. (2021) | 66.67 | 66.67 | 84.47 | 1.89 | 6.06 | 17.05 |
| ELoFTR Wang et al. (2024c) | 81.82 | 81.82 | 85.61 | 4.55 | 7.20 | 21.97 |
| ASpanFormer Chen et al. (2022) | 80.30 | 81.06 | 85.61 | 10.23 | 10.23 | 20.46 |
| DUSt3R Wang et al. (2024b) | 1.52 | 3.78 | 16.29 | 0.00 | 0.00 | 0.00 |
| MASt3R Leroy et al. (2024) | 76.14 | 76.52 | 87.50 | 0.38 | 2.65 | 7.96 |
| RoMa Edstedt et al. (2024b) | 95.83 | 95.83 | 96.59 | 34.37 | 44.32 | 62.12 |
| RoMa (GIM) Shen et al. (2024) | 94.32 | 94.32 | 97.35 | 28.79 | 43.56 | 63.26 |
| RoMa (MA) He et al. (2025) | 90.15 | 90.15 | 91.29 | 22.73 | 42.05 | 64.39 |
| **RoMa (ours)** | **97.35** | **97.35** | **98.11** | **45.46** | **53.41** | **73.11** |

RoMa (MA) He et al. (2025). Due to the large resolution gap between query and reference images, both VGGT Wang et al. (2025) and DUSt3R Wang et al. (2024b) fail to produce satisfactory results.

On the LQ map, in addition to geometric viewpoint changes, degraded image quality and long-term appearance variations further challenge image matching algorithms. As a result, most detector-based methods fail to find sufficient reliable correspondences and do not yield successful pose estimations. Similarly, the success rate of detector-free methods also drops significantly. Nevertheless, our trained RoMa remains the top-performing approach, clearly outperforming all other RoMa-based variants. This demonstrates the effectiveness of our synthetic dataset in modeling the geometric variations between UAV and nadir-view imagery, thereby substantially improving localization robustness.

It is important to emphasize that all results are obtained in a zero-shot setting: the training data comprises MegaDepth Li & Snavely (2018) and our proposed AerialExtreMatch, with no overlap with the localization test set. Qualitative localization results of RoMa-based methods on both HQ and LQ maps are shown in Figures 5 and 6.

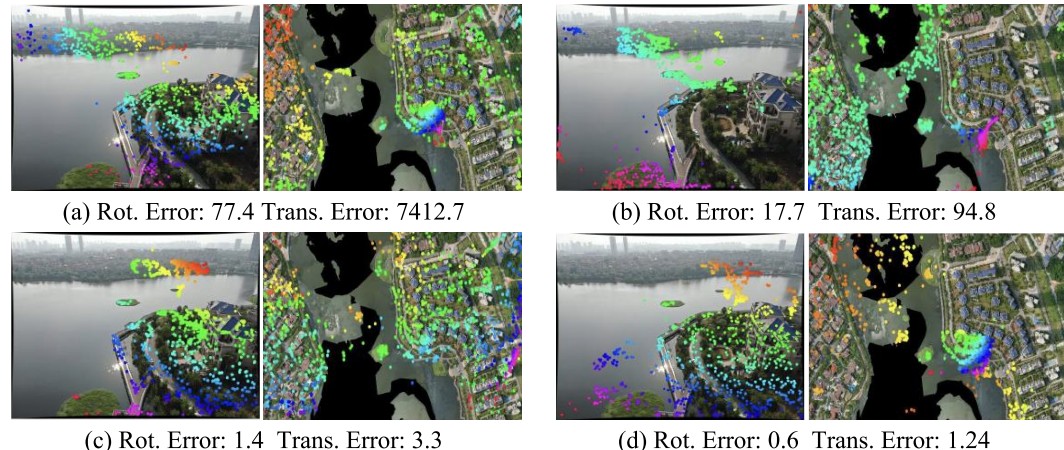

(a) Rot. Error: 77.4 Trans. Error: 7412.7

(b) Rot. Error: 17.7 Trans. Error: 94.8

(c) Rot. Error: 1.4 Trans. Error: 3.3

(d) Rot. Error: 0.6 Trans. Error: 1.24

Figure 5: **Qualitative results on the high-quality map.** Subfigures (a)–(d) show the results of the original RoMa, RoMa (MA), RoMa (GIM), and our trained RoMa, respectively.

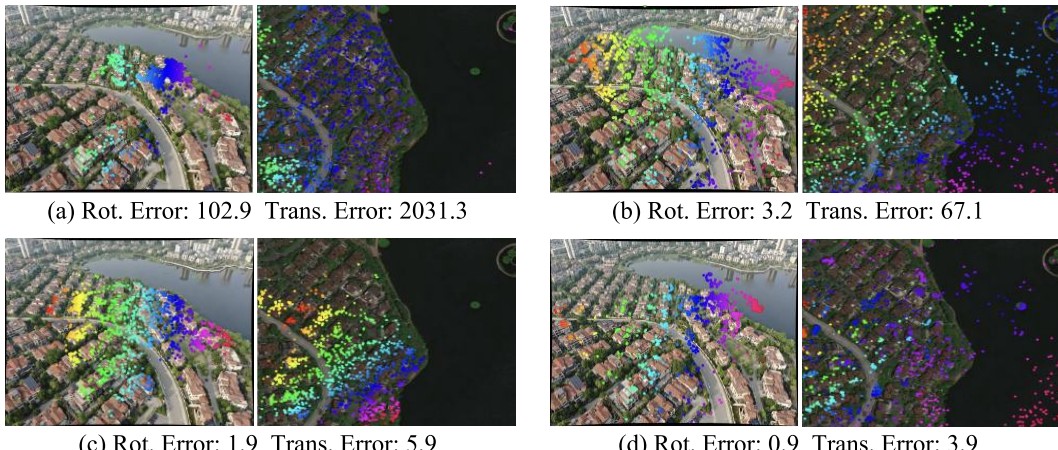

(a) Rot. Error: 102.9 Trans. Error: 2031.3

(b) Rot. Error: 3.2 Trans. Error: 67.1

(c) Rot. Error: 1.9 Trans. Error: 5.9

(d) Rot. Error: 0.9 Trans. Error: 3.9

Figure 6: **Qualitative results on the low-quality map.** Subfigures (a)–(d) show the results of the original RoMa, RoMa (MA), RoMa (GIM), and our trained RoMa, respectively.

## 5 CONCLUSION

We propose **AerialExtreMatch**, a large-scale synthetic dataset constructed from high-fidelity 3D models to simulate geometric transformations between aerial and nadir viewpoints, thereby enriching the training data for image matching tasks. In addition, we introduce the *Evaluation Pair*, a fine-grained benchmarking suite that categorizes image pairs into 32 difficulty levels based on overlap ratio, scale variation, and pitch difference, enabling a systematic evaluation of model robustness under diverse geometric configurations. To promote research in UAV localization with satellite guidance, we also provide two geo-aligned reference maps of varying reconstruction quality, supporting fair and realistic assessment of existing matching approaches. Extensive experiments show that training on AerialExtreMatch substantially improves model resilience to extreme viewpoint changes in aerial scenarios.

Despite these strengths, AerialExtreMatch has certain limitations. As it is rendered synthetically, it lacks variations in illumination and weather conditions. Additionally, challenges such as foreground occlusion are not yet considered in the current benchmark. We leave these directions for future exploration.

## ETHICS STATEMENT

Our proposed synthetic dataset is generated from open-source 3D models that are licensed for academic research use. The collected real-world UAV data does not contain personally identifiable information such as faces or license plates, and the real-world maps are restricted to academic research purposes only. We adhere to the ICLR Code of Ethics and ensure that the datasets and experiments presented in this work comply with privacy, legal, and research integrity considerations.

## REPRODUCIBILITY STATEMENT

We have made extensive efforts to ensure reproducibility. Details regarding dataset distribution and the collection process of the real UAV data are provided in the Appendix. The paper describes the training settings, evaluation protocols, and metrics in the main text, while additional implementation details are included in the supplementary material. Together, these resources enable the community to reliably reproduce our results.

## LLM USAGE

We used a large language model (LLM) solely for proofreading and language refinement of the manuscript. The LLM was not involved in research ideation, methodology design, experimental execution, or analysis of results.

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

# A APPENDIX

The following section provides additional details of AerialExtreMatch (Section A.1) and reports comprehensive experimental results on our proposed dataset (Section A.2). In addition, we present results from mixed training on other datasets, evaluations on standard ground-level benchmarks (Section A.3), and further experiments on additional UAV localization datasets (Section A.4).

## A.1 AERIALEXTREMATCH DETAILS

We introduce the sampling distribution of Train Pair in **AerialExtreMatch** in Section A.1.1. Section A.1.2 presents the criteria for defining difficulty levels, along with corresponding visual examples. Section A.1.3 describes the process of generating ground-truth poses for query images and illustrates the differences between reference maps of varying quality.

### A.1.1 TRAINING DATA DETAILS

Figure 7 illustrates the distribution of sampled regions across continents, the spatial layout within Europe, and the sampling patterns of nadir and oblique viewpoints. Nadir views are obtained at uniform spatial intervals, with three images rendered at different altitudes for each location. In contrast, oblique views are stochastically sampled to emulate low-altitude UAV perspectives within each region. Our sampling procedure adopts a three-level hierarchy: continent → country → region. For each continent, we select multiple urban areas from different countries and synthesize data using 3D assets provided by Cesium for Unreal. Since the platform offers substantially more 3D models for Europe than for other continents, European samples naturally constitute the largest portion of our dataset.

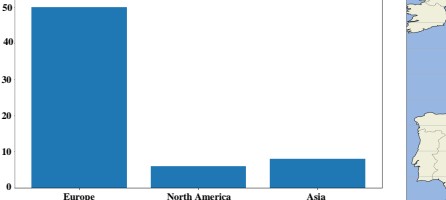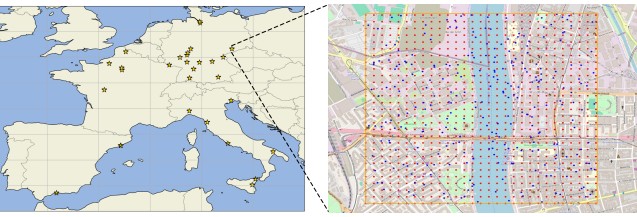

Figure 7: **Histogram and sampling examples of region distribution.** The left shows the histogram of sampling locations across continents. The middle visualizes the sampled regions within Europe, where each star denotes one sampled area. The right illustrates the viewpoint distribution within each region: points represent oblique views, and points indicate simulated nadir views.

### A.1.2 EVALUATION PAIR DETAILS

Table 3 summarizes the number of image pairs per difficulty level across different geometric variables. The evaluation set is divided into 32 levels, forming two cycles of 16 levels each. Levels 1–16 correspond to scale variations in the range of $[1.0, 2.0]$, while levels 17–32 follow the same configuration of overlap and pitch differences but with scale values exceeding 2.0. Except for levels 17 and 20, which contain 665 and 387 pairs, respectively, all other levels from 17 to 32 contain 1000 image pairs. Figure 8 presents a visualization example for each difficulty level. The top panel shows examples with scale in $[1.0, 2.0]$, and the bottom panel corresponds to examples with scale greater than 2.0. Within each panel, rows represent levels with the same overlap ratio but increasing pitch differences, while columns represent levels with the same pitch difference but decreasing overlap ratios.

### A.1.3 LOCALIZATION DETAILS

**Query Image Collect.** We collect the query images $I_q$ using a DJI M300 drone equipped with the H20T camera, which integrates three types of lenses. For our localization experiments, we utilize the wide-angle images. The wide-angle lens is calibrated indoors, and distortion-corrected images are used for all evaluations. Figure 9 shows the image acquisition device, and Table 4 lists the detailed camera parameters.

Table 3: **Difficulty level definitions.** The 32 levels are divided into two cycles of 16 levels each. Levels 17–32 follow the same variation patterns as Levels 1–16 in terms of overlap and pitch difference, but with a higher scale range.

| Level | Overlap | Pitch Difference | Scale Variation | Num. |
|-------|---------|------------------|-----------------|------|
| 1 | >60 | 55–60 | 1.0 − 2.0 | 1000 |
| 2 | >60 | 60–65 | 1.0 − 2.0 | 1000 |
| 3 | >60 | 65–70 | 1.0 − 2.0 | 1000 |
| 4 | >60 | 70–75 | 1.0 − 2.0 | 505 |
| 5 | 40–60 | 55–60 | 1.0 − 2.0 | 1000 |
| 6 | 40–60 | 60–65 | 1.0 − 2.0 | 1000 |
| 7 | 40–60 | 65–70 | 1.0 − 2.0 | 1000 |
| 8 | 40–60 | 70–75 | 1.0 − 2.0 | 1000 |
| 9 | 20–40 | 55–60 | 1.0 − 2.0 | 1000 |
| 10 | 20–40 | 60–65 | 1.0 − 2.0 | 1000 |
| 11 | 20–40 | 65–70 | 1.0 − 2.0 | 1000 |
| 12 | 20–40 | 70–75 | 1.0 − 2.0 | 1000 |
| 13 | <20 | 55–60 | 1.0 − 2.0 | 1000 |
| 14 | <20 | 60–65 | 1.0 − 2.0 | 1000 |
| 15 | <20 | 65–70 | 1.0 − 2.0 | 1000 |
| 16 | <20 | 70–75 | 1.0 − 2.0 | 1000 |

**Query Image GT Pose.** We adopt the Render2Loc Yan et al. (2023) strategy to obtain ground-truth poses $\xi_{gt}$. Specifically, for each query image $I_q$ with a known pose $\xi_{prior}$, we render an RGB-D image from a high-fidelity 3D model at the given pose. A state-of-the-art image matching method is then applied to establish 2D-2D correspondences between the query and the rendered images. With the rendered depth $I_D$, we further derive 2D-3D correspondences. Finally, a precise camera pose is estimated using a PnP solver with RANSAC. Figure 10 illustrates the matching process on two scenes with four image pairs, along with comparisons between real images and the corresponding renderings.

**Reference Map.** We provide two types of reference maps: a high-quality DOM and DSM generated from textured 3D models, and a low-quality version reconstructed from satellite imagery. Figure 11 visualizes the differences between the two map qualities. The high-quality map exhibits clear RGB textures, no long-term environmental changes, high geometric accuracy, and detailed depth maps. In contrast, the low-quality map suffers from blurry textures, long-term changes, and blocky depth maps with limited detail.

## A.2 MORE RESULTS ON OUR BENCHMARK

In the main paper, we selectively report the performance of several matching methods on the Evaluation Pair subset. Comprehensive results for all methods across different difficulty levels will be presented in Section A.2.1. Additionally, while the main paper includes localization benchmark results for all methods, more qualitative visualizations will be provided in Section A.2.2.

### A.2.1 MATCH BENCHMARK

Tables 5–8 report detailed AUC@5° results for all 16 evaluated methods. Across all difficulty levels, ALIKED Zhao et al. (2023)+LG Lindenberger et al. (2023) consistently achieves the best performance among detector-based methods. For detector-free approaches, LoFTR Sun et al. (2021) performs best under high-overlap and low-pitch-difference conditions (i.e., Levels 1 and 17), while our fine-tuned RoMa achieves superior performance across the remaining levels. DUST3R Wang et al. (2024b) is not originally designed for image matching, consistently underperforms compared to other methods. Apart from DUST3R Wang et al. (2024b), detector-free methods generally outperform detector-based ones across most difficulty settings.

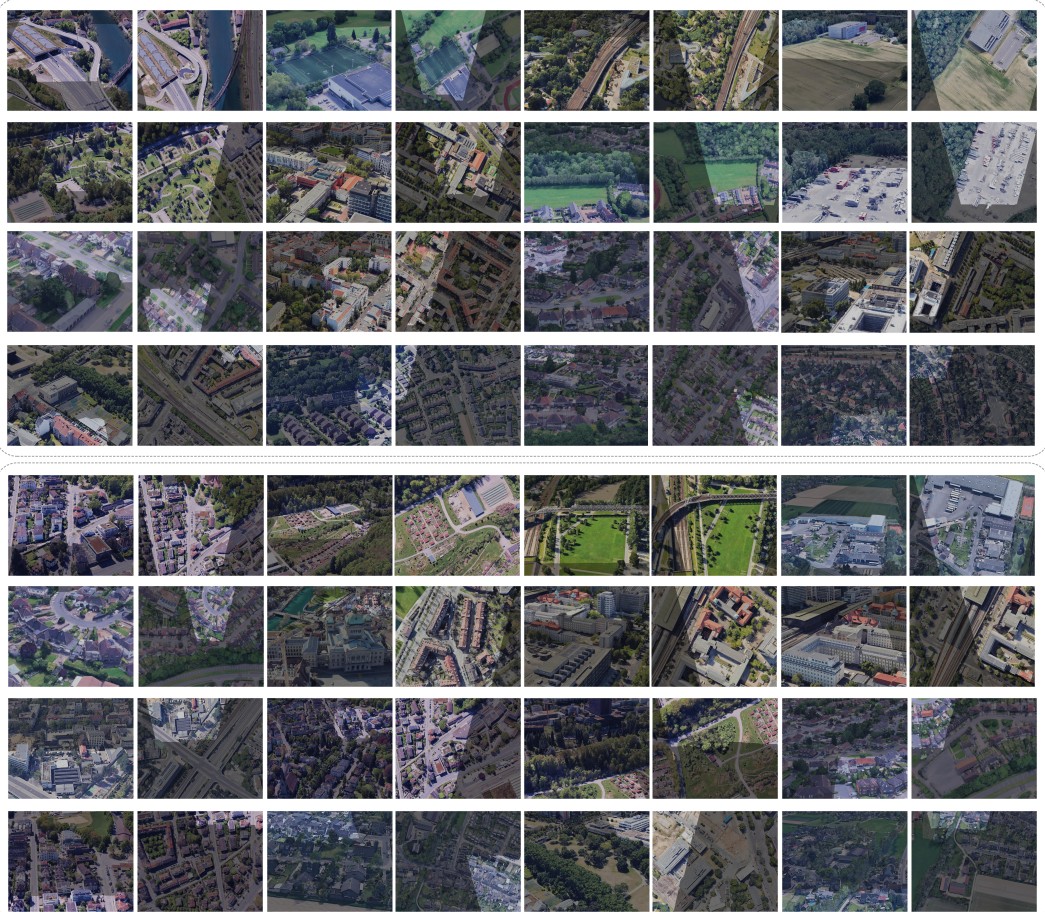

Figure 8: **Visualization of evaluation pairs across different difficulty levels.** The top panel shows levels 1–16 with scale in $[1.0, 2.0]$, while the bottom panel shows levels 17–32 with scale greater than 2.0. Each row shares the same overlap level with increasing pitch differences, and each column shares the same pitch level with decreasing overlap ratios.

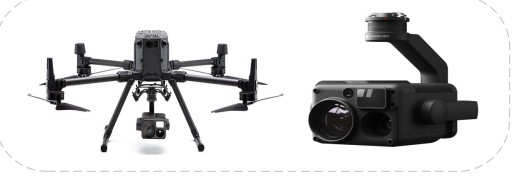

| Camera Types | Camera Parameters |
|---|---|
| | Sensor Width: 6.29 |
| | Sensor Height: 4.71 |
| Wide-angle Lens | Focal Length: 4.5 |
| | Image Resolution: $4056 \times 3040$ |

Figure 9: The capture device DJI M300 RTK mounted H20T.

Table 4: The wide-angle lens camera parameters.

Within each 4-level cycle, the overlap ratio remains fixed while the pitch difference increases, leading to a consistent performance drop for all methods. Under easy conditions (high overlap, low pitch difference), both detector-based and detector-free methods perform well. However, as the overlap decreases and pitch difference grows, detector-based methods degrade significantly. In the hardest cases (Levels 16 and 32), the best-performing detector-free method outperforms the best detector-based counterpart by a factor of three.

### A.2.2 LOCALIZATION BENCHARK

We report the localization performance of all methods in the main paper. Figure 12 and Figure 13 further present qualitative visualizations of different methods. All methods perform well on the high-

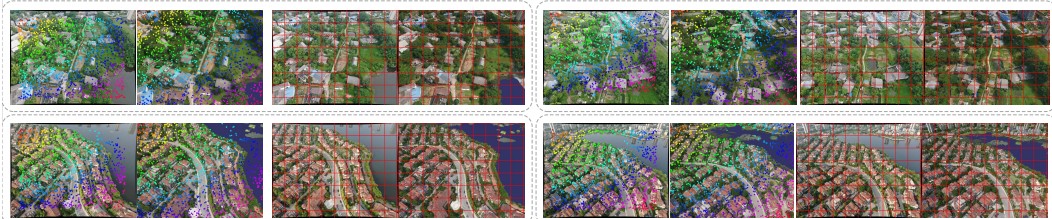

Figure 10: **Qualitative matching results and ground-truth quality visualization.** We present four comparison results from rural and urban scenes. In each panel, the left side shows matching results between the query image and the rendering from the prior pose, while the right side shows a visual comparison between the query image and the rendering from the ground-truth pose. The query image is shown on the left in both views.

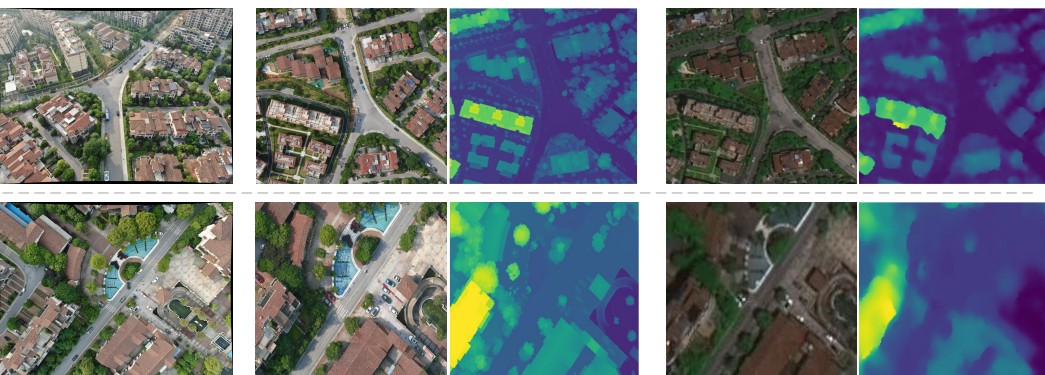

Figure 11: **Visualization of reference maps with different quality levels.** The left shows the query image, the middle displays the RGB-Depth pair rendered from the high-quality map, and the right shows the RGB-Depth pair from the low-quality map. Although the maps are geo-aligned, the low-quality version exhibits degraded appearance and reduced geometric accuracy.

quality reference map, while detector-free approaches exhibit greater robustness, with RoMa-based methods achieving the best results.

## A.3 IMPACT OF TRAINING DATA COMPOSITION

BlendedMVS Yao et al. (2020) is an open-source dataset that provides both ground-level and aerial views. To evaluate its impact, we retrain RoMa on BlendedMVS while keeping all other configurations identical, and report results on both our UAV localization benchmark and the standard MegaDepth-1500 dataset. As shown in Table 9, the left and right columns correspond to high- and low-quality maps, respectively. Entries where our dataset outperforms the BlendedMVS-trained counterpart are highlighted in bold.

Table 10 further shows that the performance drops by only 2% compared to the model trained solely on MegaDepth, which is comparable to joint training with BlendedMVS.

To further analyze the role of real and synthetic data, we compare RoMa trained on:

1. real-world MegaDepth only,

2. synthetic AerialExtreMatch only,

3. their combination.

Table 10: **Performance of different RoMa variants on MegaDepth-1500.**

| Method | 5° ↑ | 10° ↑ | 20° ↑ |
|---|---|---|---|
| RoMa (only MegaDepth) | 62.6 | 76.7 | 86.3 |
| RoMa (+BlendedMVS) | 60.9 | 75.5 | 85.7 |
| RoMa (+our data) | 60.6 | 75.3 | 85.4 |

Table 5: **AUC@5 results of different methods on Levels 1–8.**

| Method | Lv. 1 | Lv. 2 | Lv. 3 | Lv. 4 | Lv. 5 | Lv. 6 | Lv. 7 | Lv. 8 |
|---|---|---|---|---|---|---|---|---|
| ALIKED Zhao et al. (2023)+LG Lindenberger et al. (2023) | 85.97 | 81.20 | 74.81 | 71.81 | 79.72 | 62.26 | 40.82 | 25.84 |
| DISK Tyszkiewicz et al. (2020)+LG Lindenberger et al. (2023) | 84.82 | 71.53 | 61.02 | 57.67 | 72.91 | 50.27 | 31.74 | 18.56 |
| DeDoDe Edstedt et al. (2024a) | 61.06 | 57.98 | 51.60 | 46.87 | 42.58 | 32.15 | 21.97 | 11.71 |
| XFeat Potje et al. (2024) | 65.03 | 53.53 | 41.21 | 32.99 | 44.65 | 28.05 | 16.03 | 8.27 |
| XFeat* Potje et al. (2024) | 62.10 | 51.40 | 39.19 | 30.64 | 43.84 | 28.93 | 16.65 | 7.48 |
| XFeat Potje et al. (2024)+LG Lindenberger et al. (2023) | 75.12 | 64.37 | 49.48 | 47.19 | 58.53 | 38.85 | 21.81 | 12.47 |
| ALIKED Zhao et al. (2023)+VGGT Wang et al. (2025) | 34.98 | 26.72 | 19.63 | 19.98 | 22.72 | 15.57 | 10.25 | 7.90 |
| LoFTR Sun et al. (2021) | **90.10** | 87.63 | 84.89 | 83.08 | 84.64 | 74.47 | 54.99 | 39.97 |
| ELoFTR Wang et al. (2024c) | 87.58 | 83.85 | 78.31 | 76.55 | 80.92 | 66.78 | 47.35 | 31.52 |
| ASpanFormer Chen et al. (2022) | 86.79 | 80.80 | 70.10 | 64.93 | 75.16 | 56.36 | 34.94 | 20.65 |
| DUST3R Wang et al. (2024b) | 15.59 | 15.25 | 10.89 | 10.88 | 8.39 | 6.58 | 4.54 | 3.69 |
| MASt3R Leroy et al. (2024) | 83.82 | 80.17 | 77.25 | 74.95 | 76.43 | 65.72 | 48.74 | 36.04 |
| RoMa Edstedt et al. (2024b) | 89.50 | 87.56 | 86.81 | 86.91 | 87.43 | 80.03 | 64.09 | 53.45 |
| RoMa(GIM) Shen et al. (2024) | 89.55 | 87.72 | 86.34 | 87.19 | 87.79 | 80.59 | 64.21 | 53.47 |
| RoMa(MA) He et al. (2025) | 88.24 | 84.63 | 82.26 | 83.82 | 83.08 | 69.44 | 50.94 | 37.28 |
| **RoMa(ours)** | 89.88 | **88.22** | **87.61** | **88.69** | **88.28** | **83.10** | **74.80** | **68.67** |

Table 6: **AUC@5 results of different methods on Levels 9–16.**

| Method | Lv. 9 | Lv. 10 | Lv. 11 | Lv. 12 | Lv. 13 | Lv. 14 | Lv. 15 | Lv. 16 |
|---|---|---|---|---|---|---|---|---|
| ALIKED Zhao et al. (2023)+LG Lindenberger et al. (2023) | 63.33 | 46.60 | 33.92 | 24.94 | 26.99 | 23.17 | 22.33 | 22.77 |
| DISK Tyszkiewicz et al. (2020)+LG Lindenberger et al. (2023) | 55.25 | 36.88 | 24.09 | 15.50 | 20.91 | 17.49 | 14.18 | 14.29 |
| DeDoDe Edstedt et al. (2024a) | 19.22 | 13.16 | 9.02 | 7.10 | 3.53 | 2.79 | 2.51 | 3.62 |
| XFeat Potje et al. (2024) | 17.83 | 13.18 | 9.21 | 6.18 | 2.55 | 2.57 | 2.33 | 3.58 |
| XFeat* Potje et al. (2024) | 20.75 | 13.06 | 9.56 | 5.54 | 3.40 | 2.58 | 2.01 | 3.43 |
| XFeat Potje et al. (2024)+LG Lindenberger et al. (2023) | 31.83 | 22.29 | 14.86 | 10.93 | 7.49 | 6.03 | 6.20 | 6.38 |
| ALIKED Zhao et al. (2023)+VGGT Wang et al. (2025) | 8.42 | 5.80 | 4.75 | 5.02 | 0.47 | 0.61 | 0.73 | 0.79 |
| LoFTR Sun et al. (2021) | 71.30 | 58.08 | 43.65 | 32.85 | 39.16 | 33.62 | 33.23 | 34.47 |
| ELoFTR Wang et al. (2024c) | 61.32 | 46.91 | 36.94 | 25.01 | 30.91 | 25.48 | 24.52 | 26.75 |
| ASpanFormer Chen et al. (2022) | 50.01 | 34.02 | 24.16 | 17.98 | 15.71 | 12.51 | 12.40 | 11.84 |
| DUST3R Wang et al. (2024b) | 2.45 | 2.26 | 1.88 | 2.76 | 0.29 | 0.23 | 0.26 | 0.60 |
| MASt3R Leroy et al. (2024) | 57.16 | 46.61 | 34.71 | 28.52 | 20.58 | 19.14 | 18.05 | 19.41 |
| RoMa Edstedt et al. (2024b) | 79.52 | 70.69 | 59.84 | 52.70 | 56.51 | 54.78 | 55.45 | 52.17 |
| RoMa(GIM) Shen et al. (2024) | 80.24 | 68.53 | 57.04 | 49.90 | 53.46 | 51.46 | 50.19 | 51.13 |
| RoMa(MA) He et al. (2025) | 68.17 | 55.73 | 42.14 | 35.44 | 36.97 | 34.00 | 33.63 | 34.18 |
| **RoMa(ours)** | **83.57** | **78.13** | **71.47** | **70.56** | **67.03** | **67.42** | **67.85** | **68.43** |

Table 7: **AUC@5 results of different methods on Levels 17–24.**

| Method | Lv. 17 | Lv. 18 | Lv. 19 | Lv. 20 | Lv. 21 | Lv. 22 | Lv. 23 | Lv. 24 |
|---|---|---|---|---|---|---|---|---|
| ALIKED Zhao et al. (2023)+LG Lindenberger et al. (2023) | 84.92 | 76.56 | 70.87 | 73.15 | 71.04 | 44.80 | 25.10 | 24.32 |
| DISK Tyszkiewicz et al. (2020)+LG Lindenberger et al. (2023) | 83.79 | 66.72 | 56.59 | 64.04 | 64.71 | 37.08 | 19.28 | 18.64 |
| DeDoDe Edstedt et al. (2024a) | 57.58 | 59.18 | 54.01 | 53.28 | 33.38 | 23.24 | 13.65 | 12.15 |
| XFeat Potje et al. (2024) | 56.64 | 49.97 | 41.43 | 47.03 | 30.95 | 20.23 | 10.94 | 9.95 |
| XFeat* Potje et al. (2024) | 57.69 | 51.06 | 43.12 | 45.12 | 36.10 | 21.92 | 11.02 | 9.39 |
| XFeat Potje et al. (2024)+LG Lindenberger et al. (2023) | 71.59 | 56.94 | 48.56 | 52.57 | 46.64 | 25.27 | 13.07 | 12.96 |
| ALIKED Zhao et al. (2023)+VGGT Wang et al. (2025) | 33.70 | 22.30 | 18.03 | 22.32 | 20.72 | 10.00 | 5.47 | 7.24 |
| LoFTR Sun et al. (2021) | **88.64** | 84.39 | 81.25 | 83.23 | 74.43 | 56.33 | 39.91 | 39.08 |
| ELoFTR Wang et al. (2024c) | 86.52 | 78.90 | 75.50 | 78.56 | 72.30 | 51.75 | 33.37 | 31.13 |
| ASpanFormer Chen et al. (2022) | 86.32 | 76.34 | 66.40 | 67.88 | 69.49 | 41.07 | 22.65 | 21.54 |
| DUST3R Wang et al. (2024b) | 12.01 | 11.57 | 9.26 | 10.26 | 3.15 | 2.18 | 0.89 | 1.68 |
| MASt3R Leroy et al. (2024) | 82.85 | 78.89 | 74.87 | 75.80 | 72.43 | 52.32 | 36.15 | 34.13 |
| RoMa Edstedt et al. (2024b) | 88.00 | 84.34 | 82.35 | 84.55 | 85.93 | 66.71 | 51.30 | 50.60 |
| RoMa(GIM) Shen et al. (2024) | 88.55 | 84.86 | 82.55 | 84.57 | 86.59 | 66.02 | 49.16 | 47.27 |
| RoMa(MA) He et al. (2025) | 86.58 | 80.60 | 76.45 | 79.50 | 78.46 | 52.65 | 34.65 | 33.81 |
| **RoMa(ours)** | 88.25 | **85.16** | **83.32** | **85.82** | **86.97** | **72.09** | **63.03** | **63.40** |

Table 8: **AUC@5 results of different methods on Levels 25–32.**

| Method | Lv. 25 | Lv. 26 | Lv. 27 | Lv. 28 | Lv. 29 | Lv. 30 | Lv. 31 | Lv. 32 |
|---|---|---|---|---|---|---|---|---|
| ALIKED Zhao et al. (2023)+LG Lindenberger et al. (2023) | 56.42 | 38.50 | 32.22 | 30.31 | 27.49 | 27.84 | 24.67 | 23.38 |
| DISK Tyszkiewicz et al. (2020)+LG Lindenberger et al. (2023) | 47.22 | 31.14 | 24.98 | 22.44 | 20.83 | 23.99 | 18.27 | 13.58 |
| DeDoDe Edstedt et al. (2024a) | 16.32 | 11.05 | 10.52 | 8.82 | 4.00 | 3.91 | 4.42 | 3.79 |
| XFeat Potje et al. (2024) | 18.13 | 13.74 | 14.13 | 10.79 | 4.05 | 4.34 | 4.94 | 3.20 |
| XFeat* Potje et al. (2024) | 19.89 | 14.11 | 14.67 | 9.87 | 4.42 | 5.30 | 4.86 | 3.48 |
| XFeat Potje et al. (2024)+LG Lindenberger et al. (2023) | 25.88 | 18.88 | 18.51 | 14.81 | 7.79 | 9.33 | 8.14 | 7.88 |
| ALIKED Zhao et al. (2023)+VGGT Wang et al. (2025) | 8.09 | 5.08 | 5.82 | 6.94 | 0.64 | 1.11 | 0.82 | 1.06 |
| LoFTR Sun et al. (2021) | 62.81 | 49.96 | 44.67 | 41.99 | 38.47 | 41.01 | 38.49 | 34.07 |
| ELoFTR Wang et al. (2024c) | 57.64 | 41.80 | 34.73 | 33.61 | 33.03 | 32.04 | 29.19 | 26.92 |
| ASpanFormer Chen et al. (2022) | 45.62 | 28.53 | 25.33 | 21.32 | 17.56 | 16.80 | 14.65 | 12.88 |
| DUST3R Wang et al. (2024b) | 0.84 | 1.96 | 2.70 | 2.39 | 0.16 | 0.37 | 0.53 | 0.54 |
| MASt3r Leroy et al. (2024) | 54.16 | 39.93 | 34.39 | 33.13 | 21.06 | 20.12 | 19.55 | 19.99 |
| RoMa Edstedt et al. (2024b) | 74.97 | 63.08 | 56.57 | 57.33 | 55.12 | 55.22 | 53.39 | 54.53 |
| RoMa(GIM) Shen et al. (2024) | 75.55 | 60.49 | 52.94 | 53.12 | 52.95 | 52.51 | 50.26 | 48.93 |
| RoMa(MA) He et al. (2025) | 62.42 | 46.55 | 41.07 | 38.52 | 37.03 | 36.17 | 35.47 | 32.78 |
| **RoMa(ours)** | **79.86** | **69.61** | **67.43** | **70.83** | **64.09** | **65.46** | **67.14** | **68.48** |

Table 9: **Performance comparison on AerialExtreLocalization with different training datasets.** .

| Method | High-quality map | | | Low-quality map | | |
|---|---|---|---|---|---|---|
| | (5m, 1°) | (10m, 1°) | (20m, 2°) | (5m, 1°) | (10m, 1°) | (20m, 2°) |
| RoMa (MegaDepth + BlendedMVS) | 91.67 | 91.67 | 94.32 | 25.76 | 32.96 | 53.03 |
| RoMa (MegaDepth + AerialExtreMatch) | **97.35** | **97.35** | **98.11** | **45.46** | **53.41** | **73.11** |

Table 11: **Performance comparison on AerialExtreLocalization with different training data compositions.**

| Method | High-quality map | | | Low-quality map | | |
|---|---|---|---|---|---|---|
| | (5m, 1°) | (10m, 1°) | (20m, 2°) | (5m, 1°) | (10m, 1°) | (20m, 2°) |
| RoMa (only MegaDepth) | 95.83 | 95.83 | 96.59 | 34.37 | 44.32 | 62.12 |
| RoMa (AerialExtreMatch) | 89.77 | 89.77 | 90.53 | 9.09 | 10.61 | 27.65 |
| RoMa (MegaDepth + AerialExtreMatch) | 97.35 | 97.35 | 98.11 | 45.46 | 53.41 | 73.11 |

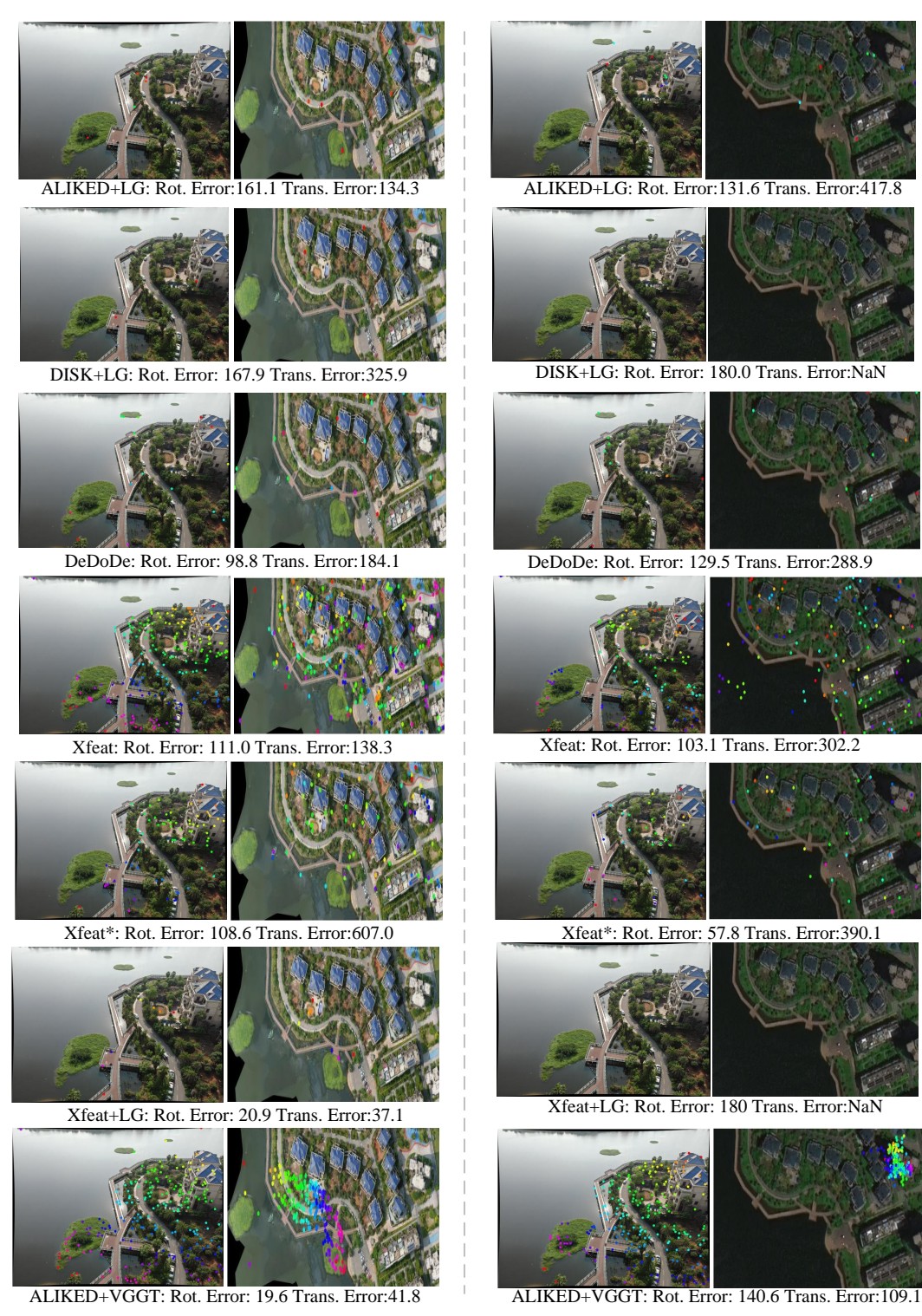

Figure 12: **Qualitative results of detector-based methods on reference maps of different quality.**
Left: results on the high-quality map. Right: results on the low-quality map.

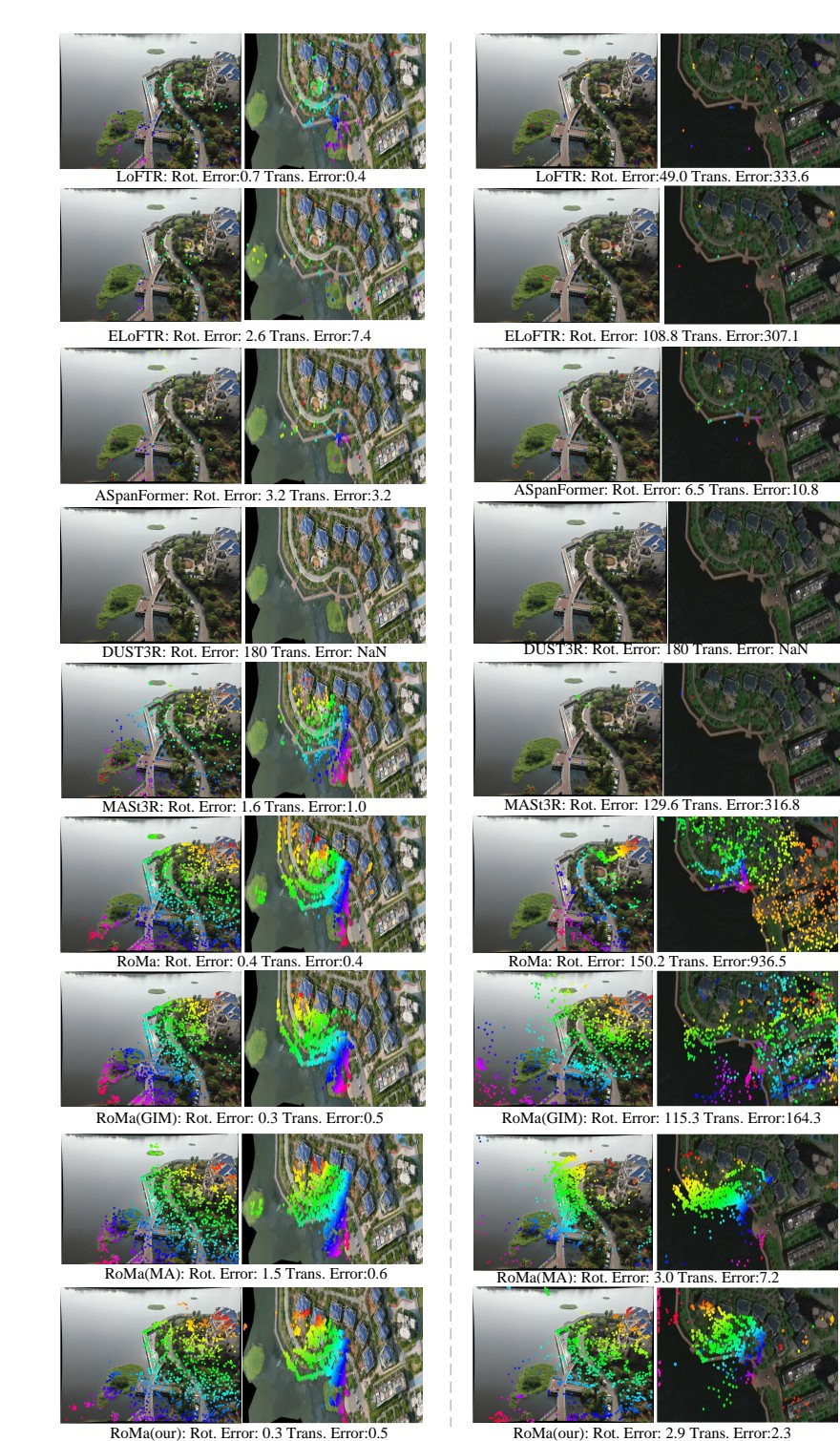

Figure 13: **Qualitative results of detector-free methods on reference maps of different quality.**
Left: results on the high-quality map. Right: results on the low-quality map.

Table 11 reports results on real-world localization maps of different quality. The findings suggest that real data is essential for strong generalization, while synthetic data contributes complementary scale and diversity.

### A.4 MORE RESULTS ON OTHER LOCALIZATION DATASETS

Both UAVD4L Wu et al. (2024) and UAVVisLoc Xu et al. (2024) are real-world UAV localization datasets. We evaluate our trained RoMa against its variants and other competing methods. RoMa trained with our proposed dataset consistently outperforms other RoMa variants, as reported in Table 12. UAVD4L provides full 6DoF UAV poses, and we report recall with respect to both translation and rotation. In contrast, UAVVisLoc only provides ground-truth translation, and thus, evaluation is limited to translation recall.

Table 12: **Visual localization results on UAVD4L and UAVVisLoc.** RoMa trained with our proposed dataset achieves slightly higher performance compared to the original RoMa.

| Method | UAVD4L | | | UAVVisLoc | | | |
|---|---|---|---|---|---|---|---|
| | (5m, 1°) | (10m, 1°) | (20m, 2°) | median ↓ | acc@5m | acc@10m | acc@20m |
| ELoFTR Wang et al. (2024c) | 64.00 | 64.67 | 97.33 | 2.92 | 60 | 84 | 92 |
| DUSt3R Wang et al. (2024b) | 54.67 | 54.67 | 94.00 | 7.31 | 28 | 64 | 84 |
| MASt3R Leroy et al. (2024) | 60.67 | 60.67 | 97.33 | 4.29 | 56 | 68 | 76 |
| RoMa Edstedt et al. (2024b) | 65.33 | 65.33 | 97.33 | 2.55 | 84 | 92 | 100 |
| RoMa (GIM) Shen et al. (2024) | 68.67 | 68.67 | 97.33 | 2.47 | 84 | 92 | 100 |
| RoMa (MA) He et al. (2025) | 68.00 | 68.00 | 97.33 | 2.53 | 84 | 88 | 96 |
| RoMa (our) | **68.67** | **68.67** | **97.33** | **2.28** | **84** | **92** | **100** |

