# OpenReview forum: "AerialExtreMatch: A Benchmark for Extreme-View Image Matching and Localization"
_ICLR.cc/2026/Conference — ICLR 2026 Conference Withdrawn Submission_

### Official Review · Reviewer_hhiD · 2025-10-29

**Soundness:** 3
**Presentation:** 3
**Contribution:** 2
**Rating:** 6
**Confidence:** 4

**Summary:**

This paper introduces AerialExtreMatch, a large-scale synthetic dataset designed for extreme-view image matching between UAV and satellite imagery, a critical task for UAV localization. To address the lack of real-world data covering severe geometric variations, the authors create ~1.5 million synthetic image pairs from 3D scenes. A key contribution is a hierarchical benchmark with 32 difficulty levels based on overlap, scale, and pitch difference. Experiments on 16 methods show that training on this dataset yields substantial performance gains in both image matching and real-world UAV localization tasks.

**Strengths:**

The introduced AerialExtreMatch dataset is valuable. Its ~1.5 million synthetic pairs, explicitly tailored to the challenging problem of UAV-satellite matching with extreme geometric variations, provide a foundational resource that was previously lacking, enabling robust training and nuanced evaluation.

The hierarchical benchmarking with 32 difficulty levels enables a fine-grained analysis of model performance that was not previously possible.

Comprehensive experimental validation demonstrates clear and substantial performance gains across a wide range of existing methods.

**Weaknesses:**

1. The paper's contribution is primarily the dataset and benchmark. While valuable, it does not introduce a novel algorithmic method. This may raise questions about its suitability for a top-tier conference like ICLR, which often prioritizes methodological innovation.

2. The writing lacks clarity in several key areas, leading to significant questions about the dataset's design choices and practical validity. Please refer to my comments in "Questions".

**Questions:**

1. The training set includes approximately 1.5M pairs with variations in flight altitude and pitch angles. Why exclude considerations of variations in roll and yaw angles?

2. The proposed dataset includes both synthetic and real image pairs. How do these constitute the training and testing set? Are only synthetic image pairs used for training and only real-world image pairs for testing, or is there a mixture, and if so, what kind of mixture? Is there any demonstration that the proposed 1.5M synthetic image pairs can facilitate real-world UAV localization based on satellite images?

3. It is better to add real & synthetic descriptions in Tab. 1.

4. How are the variations in visual quality among DSMs and DOMs rendered from high-quality textured 3D models as well as reconstructions derived from satellite images? How is the reconstruction from satellite images performed?

5. The pitch angles are chosen between 50$^\circ$ and 75$^\circ$. Why use this range? Why not consider angles lower than 50$^\circ$ or larger than 75$^\circ$? Any practical accordance?

6. Synthetic satellite images are generated at three heights -- 300m, 400m, and 500m. What are the ground resolutions respectively? Each RGB-depth pair is rendered at a resolution of 1280 x 1024. This refers to UAV image and its corresponding depth, or satellite image and its corresponding depth?

7. How is the synthetic UAV & satellite image pair constructed, the same as the construction method for real images illustrated in Line 315 (lack equation labels)? I understand this method is good for reconstructing an image pair with sufficient overlaps. However, in practical UAV localization, how to determine the coverage of the satellite image for each UAV image, since the 3D points of the satellite images and the pose of the UAV images are unavailable?

8. Among the comparison algorithms, does the author only train RoMa on the proposed dataset, while using the pre-trained weights of other methods from their original dataset? If so, the performance gain on RoMa is not really significant compared to the the counterpart version RoMa (GIM) on HQ images. This raises another question: why not compare RoMa (ours) with VGGT and π^3[*]?

[*] Wang et al. π^3: Permutation-Equivariant Visual Geometry Learning

I do not intend to say that this paper did not compare to xxx and thus this is a weakness. Rather, my question is: since the contribution of this paper is only a dataset, is it really valuable? By fine-tuning/re-training on the proposed dataset, can the proposed method outperform the most recent state-of-the-art?

---

### Official Review · Reviewer_QXT7 · 2025-11-01

**Soundness:** 2
**Presentation:** 3
**Contribution:** 3
**Rating:** 6
**Confidence:** 4

**Summary:**

This paper addresses the critical challenge of matching images between Unmanned Aerial Vehicles (UAVs) and satellites, a task complicated by extreme viewpoint disparities. This work introduces AerialExtreMatch, a large-scale synthetic dataset specifically designed for extreme-view image matching and UAV localization. This dataset comprises approximately 1.5 million image pairs rendered from high-quality 3D scenes, simulating diverse UAV and satellite perspectives. It is structured with a hierarchical evaluation benchmark consisting of 32 distinct difficulty levels. Furthermore, a real-world UAV localization dataset was compiled, which includes georeferenced maps of varying visual quality. Experimental results demonstrate that models trained on AerialExtreMatch achieve significant performance gains in both extreme-view image matching and real-world localization tasks.

**Strengths:**

1. The paper constructs a comprehensive three-component benchmark: 1.5 million training image pairs, a 32-level hierarchical difficulty evaluation system, and a real-world localization dataset. This multi-level design supports both model training and fine-grained performance analysis. The difficulty ranking is based on three geometric criteria—overlap ratio, scale change, and pitch angle difference—providing a standardized framework for algorithm evaluation.
2. By leveraging high-quality 3D models from Cesium for Unreal and combining them with Unreal Engine 5 and AirSim for realistic rendering, we effectively address the high costs associated with acquiring real-world data. Synthetic data not only offers significant cost savings but also enables precise control over data generation conditions, making it possible to systematically investigate the impact of geometric transformations on matching performance.
3. The paper accurately identifies the core challenge in drone visual localization—the extreme geometric disparity between the oblique perspective of low-altitude drones and the orthographic perspective of high-altitude satellites. This issue holds significant practical importance for tasks such as emergency rescue and large-scale scene reconstruction.

**Weaknesses:**

1.Although synthetic data is low-cost and highly controllable, it lacks real-world factors such as lighting variations, weather conditions, and seasonal changes—elements that significantly impact image matching performance in practical applications. It is recommended to incorporate data augmentation strategies to simulate these variations.
2.The experiment primarily compares with general image matching methods, lacking comparisons with the latest methods specifically designed for aerial image matching. It is recommended to add comparative experiments with specialized and state-of-the-art methods in the field of aerial image processing.
3.Lack of detailed analysis of computational efficiency metrics (FLOPs, memory usage)

**Questions:**

1.How to address the domain adaptation problem from synthetic to real-world data?
2.What is the potential for compressing and optimizing models on resource-constrained devices?
3.Has the impact of dynamic objects on positioning accuracy in real-world scenarios been evaluated?

---

### Official Review · Reviewer_bmei · 2025-11-04

**Soundness:** 2
**Presentation:** 3
**Contribution:** 2
**Rating:** 2
**Confidence:** 4

**Summary:**

This paper proposes AerialExtreMatch, a large-scale synthetic benchmark aimed at addressing the challenges of extreme-view image matching and UAV localization. The authors develop a dataset with approximately 1.5 million RGB-Depth image pairs, generated from high-fidelity 3D models using Unreal Engine 5 and AirSim.
The benchmark introduces a hierarchical difficulty grading system based on overlap ratio, pitch difference, and scale variation. The authors also include a real-world UAV localization task with geo-aligned reference maps for validation. The experimental results on 16 image matching methods show that models trained on the AerialExtreMatch dataset outperform existing baselines, especially under extreme geometric variations.

**Strengths:**

Novel Dataset: The introduction of AerialExtreMatch represents a significant contribution toward bridging a critical gap in cross-view UAV–satellite image matching under extreme viewpoint variations.

Comprehensive Evaluation: Featuring a 32-level difficulty grading system, the dataset offers a structured and fine-grained evaluation framework—marking a valuable addition to the existing literature.

Synthetic Data Quality: Leveraging a robust generation pipeline with high-fidelity 3D models, the dataset ensures both visual realism and scalability, providing an effective large-scale solution for complex image matching challenges.

**Weaknesses:**

1)	While the overall language is understandable, the presentation of some critical components lacks clarity and motivation.
a)	The computation of the co-visibility mask is described procedurally but lacks a clear rationale. The potential limitations of this specific method, particularly for handling occlusions which are prevalent in extreme view changes between nadir and oblique views, are not discussed. Since this mask underpins the three proposed evaluation metrics, a more thorough explanation and justification are necessary.
b)	The spatial relationship between the training and evaluation splits is unclear. Do they share scenes or geographic locations? Potential overlaps could significantly impact the validity of the experimental results, especially given the synthetic nature of the data.
2)	Insufficient Comparison with Relevant Prior Work: The paper's primary contribution is a dataset for cross-view matching and localization. However, the comparison with existing relevant datasets is incomplete. Key datasets in the domain of UAV oblique view to satellite image matching and localization, such as University-1652, SUES-200, and Game4Loc, are not discussed or compared. Highlighting the differences and potential advantages of AerialExtreMatch against these established benchmarks is crucial.
Furthermore, evaluating methods specifically designed for or benchmarked on these related datasets on AerialExtreMatch, or using those datasets to test the generalizability of models trained on AerialExtreMatch, would significantly strengthen the validation.
3)	Limitations in Experimentals: A major concern is the fairness in the matching benchmark evaluation. The RoMa model is fine-tuned on the AerialExtreMatch training set, while other competing methods are evaluated in a zero-shot manner without similar adaptation. This puts other methods at a significant disadvantage, especially if there is any distribution shift or specific bias in the synthetic data. The potential overlap between training and evaluation data (as mentioned in Weakness 1b) further compounds this issue.
The experimental analysis feels somewhat weak. For instance, there is no detailed ablation study analyzing the individual impact of the core difficulty factors (overlap, pitch, scale) on model performance. Similarly, the localization evaluation focuses heavily on map quality but does not systematically analyze performance variation with respect to other key dataset characteristics like altitude or overlap ratio, which are central to the paper's contribution.
4)	Dataset Scope and Potential for Broader Impact: Given the synthetic nature of the primary data, increasing scene diversity in the final release would be beneficial. Furthermore, providing pixel-level correspondence ground truth, if feasible, would greatly enhance the dataset's utility for the research community beyond pose estimation tasks.

**Questions:**

1）The co-visibility mask calculation relies on a depth reprojection error threshold. Could you discuss the rationale behind this specific design choice and its potential failure modes, especially concerning occlusions common in extreme viewpoint changes?
2）Could you clarify the geographical and scene distribution relationship between the Train Pair and Evaluation Pair splits? Are they strictly disjoint in terms of 3D scenes and locations? This is critical for assessing the benchmark's validity.
3）The experimental setup involves fine-tuning RoMa on your training set while evaluating other methods zero-shot. How do you justify this as a fair comparison for the image matching benchmark? Would you consider providing results for other strong baselines (e.g., LoFTR) after fine-tuning them on AerialExtreMatch for a more equitable assessment?
4）Given the existence of other cross-view geo-localization datasets (e.g., University-1652), how does AerialExtreMatch specifically advance the field? Could you include comparisons or cross-dataset evaluations to better position your contribution relative to this related work?
5）Are there plans to expand the real-world benchmark to include a wider variety of environmental conditions or map sources? Furthermore, will pixel-level ground-truth correspondences be provided for the synthetic pairs to facilitate research on dense matching?

---

### Official Review · Reviewer_W7y9 · 2025-11-05

**Soundness:** 3
**Presentation:** 3
**Contribution:** 2
**Rating:** 2
**Confidence:** 4

**Summary:**

The paper introduces AerialExtreMatch, a benchmark targeting extreme-view image matching for UAV localization against nadir-view maps. It contains ~1.5M synthetic image pairs rendered from high-fidelity 3D scenes (Cesium/UE5/AirSim), and a graded evaluation with 32 difficulty levels defined by overlap ratio, pitch difference, and scale. The authors also build a real-world localization benchmark with geo-aligned high-quality (UAV-reconstructed) and low-quality (satellite-derived) maps and evaluate 16–17 representative methods, showing that training RoMa on AerialExtreMatch substantially improves matching and localization under large viewpoint gaps.

**Strengths:**

- Clear problem and strong motivation: Matching oblique, low-altitude UAV images to nadir maps is a real pain point; existing datasets don’t cover the geometry extremes well.
- Scale and structure: The dataset is large (≈1.5M pairs) and the 32-level difficulty design provides fine-grained analysis (overlap, pitch, scale).
- Thorough evaluation: Broad coverage of detector-based and detector-free matchers; RoMa trained on the new data improves substantially on hard cases and tops the real-world localization benchmark.

**Weaknesses:**

- Limited novelty: The core contribution is a well-engineered dataset/benchmark and training a known matcher (RoMa) on it. The algorithmic novelty is minimal; much of the value is in data creation and evaluation protocol.
- Synthetic bias and coverage:
    - The graded design controls pitch, overlap, scale, but roll and yaw are not explicit axes in the evaluation protocol; this limits robustness claims for full 6-DoF viewpoint gaps.
    - The authors themselves note no illumination/weather variations and limited modeling of occlusion, important for aerial ops (seasonality, shadows, fog, precipitation).

**Questions:**

- Can you detail the scene split between Train Pair and Evaluation Pair to rule out content leakage (same city/tiles/assets)? Any cross-renderer test (e.g., different asset providers) to measure robustness?
- For the HQ/LQ localization benchmark, some baselines fail due to large resolution gaps. Did you re-scale reference crops or adapt matcher resolutions per method to remove trivial disadvantages?

---

### Note · Authors · 2025-11-21

I have read and agree with the venue's withdrawal policy on behalf of myself and my co-authors.